# The Lived Experiences of Stigmatization in the Process of HIV Status Disclosure among People Living with HIV in Taiwan

**DOI:** 10.3390/ijerph18105089

**Published:** 2021-05-11

**Authors:** Chia-Hui Yu, Chu-Yu Huang, Nai-Ying Ko, Heng-Hsin Tung, Hui-Man Huang, Su-Fen Cheng

**Affiliations:** 1School of Nursing, College of Medicine, Chung Shan Medical University, Taichung 40201, Taiwan; cshp184@csh.org.tw; 2Center of Infection Control, Chung Shan Medical University Hospital, Taichung 40201, Taiwan; 3School of Nursing, Cedarville University, Cedarville, OH 45314, USA; huangc@cedarville.edu; 4Departments of Nursing, College of Medicine, National Cheng Kung University, Tainan 70101, Taiwan; nyko@mail.ncku.edu.tw; 5College of Nursing, National Yang Ming Chiao Tung University, Taipei 11219, Taiwan; Shannontung719@gmail.com; 6School of Nursing, Tzu Chi University of Science and Technology, Hua-lien 970302, Taiwan; hhuiman@ems.tcust.edu.tw; 7Department of Allied Health Education and Digital Learning, National Taipei University of Nursing and Health Sciences, Taipei 11219, Taiwan

**Keywords:** PLWH, HIV stigma, content analysis, HIV disclosure

## Abstract

People living with HIV (PLWH) face social stigma which makes disclosure of HIV status difficult. The purpose of this descriptive qualitative study was to understand the lived experiences of stigmatization in the process of disease disclosure among PLWH in Taiwan. Analysis of the semi-structured interviews from 19 PLWH in Taiwan revealed two phases and six themes. Phase one “experiences before disclosure” involved three themes: “Struggles under the pressure of concealing the HIV Status”, “Torn between fear of unemployment/isolation and desire to protect closed ones”, and “Being forced to disclose the HIV status.” Phase two “experiences after disclosure” included three themes: “Receiving special considerations and requirements from school or work”, “Receiving differential treatments in life and when seeking medical care”, and “Stress relief and restart.” Healthcare professionals need to assess stigmatization in PLWH and develop individualized approaches to assist with the disease disclosure process.

## 1. Introduction

Approximately 37.9 million individuals were infected by the human immunodeficiency virus (HIV) in 2018 globally [1]. In Taiwan, the current population of PLWH is 42,398 [2] with a growth rate of 195 times from 1984 to 2019. As of April 2020, 80.82% of the people living with HIV (PLWH) contracted HIV from sex, of which men who have sex with men (MSM) account for 65.1%. MSM has always been the main population of HIV infection [2]. HIV infection is a notifiable infectious disease that is typically treated with HIV combination antiretroviral therapy (cART). The current policy recommends initiation of medication treatment following the HIV diagnosis. When the medication treatment regimen is adhered to, PLWH show a similar life expectancy to the general population [3]. However, PLWH may not discontinue the cART once the treatment begins, which results in a heavy burden on medical insurance and high treatment cost (USD 440/month) [4]. Currently, the two major challenges facing HIV treatment in Taiwan are the increasing prevalence of HIV infection in the younger population and the younger population as the main HIV-infected group [2]. Once diagnosed, the young PLWH must receive the medication treatment for life. However, PLWH may have poor medication adherence and reduced willingness to seek medication attention due to discrimination and HIV-related stigma [5]. Not only does HIV stigma influence medication adherence, but it also plays a vital part in PLWH’s decisions about disclosing their HIV status to family members, friends, and healthcare professionals. They weigh the risks and benefits as they contemplate the decision to disclose the HIV status. The consequences of the disclosure might involve both positive and negative psychological and physiological outcomes such as depression, perceived social support, and negative stigmatizing responses [6].

Confucianism is the core philosophical foundation of the Taiwanese culture, which emphasizes social harmony, obligations, interdependence, and realization of social roles [7,8,9]. The philosophy values a person’s moral character and interpersonal relationships. The responsibility of procreation within a husband–wife marital relationship is emphasized [10,11]. A common Taiwanese cultural proverb states “Among the three ways of violating filial piety, having no heir is the most dishonoring.” This proverb exemplifies the cultural expectation of procreation in males. The MSM are perceived as unable to get married and having offspring. Therefore, they become less accepted by the public in Taiwan. The public perceives HIV infection as a sexually transmitted disease resulting from immoral behaviors and views PLWH as sinners (e.g., immorality in the previous life, sexual affairs, homosexual males, and immoral females). Additionally, homosexuality is typically equated with HIV infection. Therefore, placing the HIV stigmatization on an already marginalized gay population causes PLWH to become reluctant to seek medical care [12,13]. In addition to social stigmatization and discrimination, PLWH also bear shame and guilt as their family members and friends become aware of the diagnosis [14]. Stigmatization toward the HIV-positive MSM takes diverse forms and involves extensive aspects, such as discrimination and rejection by HIV-negative MSM, ageism, changes in physical appearance, and self-stigma [15].

Disclosure of HIV status is a process that involves careful consideration of the potential benefits and harm [16,17]. A study conducted in Taiwan discovered that the barriers preventing PLWH to disclose the HIV status to family were “unpredictable reactions from parents” and “social stereotype regarding HIV infection [18]”. Tsai, Lu, Wu, and Feng also found the greatest stress that PLWH in northeast China experienced was “concerns about disease being exposed” and “disease disclosure”. Disclosure of HIV status is a conflicting and difficult decision for PLWH [19]. Qiao, Li, Zhou, Shen, and Tang also discovered that 51% of the PLWH in Guangxi, China informed their family members of their HIV status [20]. Results from the 2017 Taiwan HIV Stigma and Discrimination Survey showed that the top five reasons for PLWH’s fear toward HIV screening were: fear of being isolated from family members and friends (58%), people perceived them as men having sex with men or prostitutes (52.4%), unemployment or being suspended from schools (31.6%), partner/spouse breaking up with them (29.9%), and being forced to leave home or social media groups (24.5%) [21]. Chen, Lew-Ting, Tsai, and Hsiung pointed out that PLWH faced challenges from the illness and sexual identity. They frequently experienced uneasiness and difficulties in social interactions and had preconceived beliefs about discrimination and social isolation [22]. Hutson et al. found that the most frequently reported stigma concern was disclosure of HIV infection [23]. A significantly negative correlation was found between the overall spiritual well-being and personalized stigma.

The concept of “saving face” is deeply embodied in Chinese culture. The Chinese concept of “saving face” is viewed as “a physical, emotional, social, and moral process” that emphasizes collective identity and filial piety [24]. Disrespectful actions or conduct might cause a person to lose face, which signifies shame and embarrassment for the individual and the entire family. In the Chinese culture, PLWH are perceived as “losing face” because of the preconceived moral notions and judgments on the causes of HIV/AIDS. Culturally, HIV infection is considered as a condition that brings shame to the entire family. The impact resulting from external stigma and self-stigma (e.g., shame, self-blame) negatively influences PLWH’s spiritual well-being and interpersonal relationships [25,26,27], medication adherence, and willingness to seek medical attention [5,15,28]. If the PLWH chooses to conceal the HIV infection, it would increase the burden on medical resources [29,30].

In Taiwan, the Confucius philosophical beliefs and Chinese cultural notion of morality may impact the PLWH’s HIV status disclosure. Their processes of disclosure may distinctly differ from the disclosure processes of the PLWH in Western culture. However, only limited studies have addressed the processes of HIV status disclosure among PLWH in Taiwan. Therefore, the purpose of this study was to understand the lived experiences of stigmatization in the process of HIV status disclosure among PLWH in Taiwan.

## 2. Materials and Methods

This qualitative study used phenomenological methodology to understand the meaning of the phenomena. Phenomenology is an appropriate methodology for exploring lived experiences [31,32,33].

### 2.1. Participants

A purposive sampling method was used to obtain participants from the infectious disease outpatient clinic and patient units at a hospital designated for HIV treatment in Taiwan from January to October 2016. The inclusion criteria were: (1) confirmed diagnosis of HIV-positive, and (2) ≥18-year-old. Individuals with psychiatric disorders were excluded. Nineteen PLWH participated in semi-structured interviews. None of the participants withdrew from the study.

### 2.2. Data Collection and Setting

One-on-one audio-recorded in-depth interviews were conducted using a semi-structured interview guide (Table 1). The interviews were conducted in a private room at the outpatient clinic. The first author of this study conducted all the interviews to ensure consistency in data collection. Prior to the interviews, the attending physician explained the study purpose to the potential participants then referred them to the first author. The first author obtained the informed consent and conducted the interviews. The recruitment process helped the participants to establish a trusting relationship with the researchers. Recruitment of participants continued until data saturation. The length of the interviews ranged from 30 to 60 min.

### 2.3. Data Analysis

Data were analyzed using content analysis [34]. A trained research assistant conducted verbatim transcriptions of the recorded interviews within 24 to 72 h upon completion of each interview. The researchers then thoroughly read through the transcripts to gain an understanding of the participants’ experiences of HIV status disclosure and stigmatization, and to determine content that is meaningful and relevant to the study. Similar concepts and meanings were synthesized and combined into units. Units with similar characteristics and relevance were categorized to form conceptual themes. The interviews were documented on an A4-size paper with vertical texts in two columns (left column: interview documentation, right column: analysis).

### 2.4. Methodological Rigor

Lincoln and Guba’s (1985) evaluative criteria for trustworthiness (credibility, transferability, dependability, and confirmability) were used to ensure methodological rigor [35]. The content was transcribed verbatim. To establish credibility, the authors took the following measures. The first author of this study conducted all the interviews. Each interview was audio-recorded with the participant’s permission. If questions arose, the researchers would contact the participants and conduct a second interview if needed. To ascertain transferability, the authors used open-ended questions to facilitate free expressions of perceptions and offered opportunities for the PLWH to clarify their responses as needed. To establish dependability, the author ensured accuracy of the verbatim transcriptions, analyzed the data, and engaged in peer debriefing. To ensure conformability, the author reduced bias by keeping a non-judgmental attitude during the research process and clarifying things with the PLWH as necessary.

### 2.5. Ethical Considerations

The Institutional Review Board approval was secured prior to data collection (approval number: CS2-15099). During the informed consent process, the researchers explained the study purpose and process to the potential participants. The participants were informed of the voluntary and anonymous nature of the study before completing the informed consent form. Participation was voluntary and participants had the freedom to withdrawal at any time.

## 3. Results

### 3.1. Characteristics of Participants

Nineteen PLWH participated in the semi-structured interviews. All participants were unmarried biological males aged between 22 and 42 years old (average age: 29.05 years old). Eighteen participants were gay, whereas one participant was bi-sexual. At the time of interviews, nine participants (47.37%) had regular sexual partners. Nine participants (47.37%) held at least a bachelor’s degree. Seven participants (36.84%) were living with their family members, whereas 12 participants (63.16%) were not living with family members. In 68.42% (*n* = 13) of the participants, their family members were aware of the HIV infection. In 42.11% (*n* = 8) of the participants, their friends were aware of the HIV status. The participants disclosed the HIV status to partners (88.89%), friends (75%), and family members (61.54%).

### 3.2. Themes

Analysis of the participants’ experiences of HIV status disclosure resulted in two phases and six themes (Figure 1). Not all PLWH experienced both phases (experiences before disclosure, experiences after disclosure). Some participants never disclosed their HIV status.

#### 3.2.1. Experiences before Disclosure

This phase encompassed three themes, as described below:

##### Struggles under the Pressure of Concealing the HIV Status

HIV status disclosure was a complex and difficult process for PLWH. They hesitated to disclose the illness and repeatedly considered who should know, how to tell, and when to disclose the illness. They might purposefully conceal the diagnosis (e.g., preventing others from knowing that they have been taking medications) and use various methods to detect their family and friends’ views on people with HIV infection. The men and their families experienced varying degrees of stress because they concealed the illness from families or asked family members to assist with concealing the illness. Therefore, some PLWH would disclose the illness to their families and friends in order to decrease the stress. The stress resulting from the recurring back and forth consideration about HIV status disclosure often created strong apprehension and restlessness. The participants described this experience as follows:

*“I do not take the antiretroviral medication in front of my colleagues. If I need to take my medication, I would pretend to drink water. So, my colleagues would not see it. Or, I would take my medication at a place where my colleagues would not see, like the lunch room. Sometimes, my colleagues would ask me “do you have cold?”. I would tell them “yes” or I was just taking B complex. I have been hiding it. I do not voluntarily tell them my disease”*.(participant #3, 27 years old)

*“Initially,**only my older sister knew about my diagnosis. However, I sensed that this has created stress on her. I am the only son and, therefore, bear the responsibility to have children to carry the family name. However, I am a gay person with HIV infection. My parents would be unable to accept it if they found out about my diagnosis. I come from a traditional family. My parents have great expectations for me. So, after my older sister became aware of my illness, I asked her not to tell anyone. But, I could tell that this secret has created a lot of stress on her. I have been thinking if I should go ahead and tell my parents”*.(participant #6, 25 years old)

##### Torn between Fear of Unemployment/Isolation and Desire to Protect Closed Ones

The participants indicated that the general public perceived PLWH as dirty and disgusting and the disease as very contagious. Therefore, when they detected the misconceptions and criticisms from the public and social media, they became more concerned about unemployment and being excluded from their family circles and workplaces. On the other hand, they also wanted to protect friends, family, partners, and healthcare providers from HIV infection. Their experiences are presented below:

*“My sister and I watched a TV show a few years ago. When she knew about the actor’s HIV-positive status, she had a strong reaction. She said who dared to bury their corpses! This incident reminded me that my sister cannot accept HIV-positive people. This is why I would never talk about my diagnosis”*.(participant #10, 31 years old)

*“I choose to tell my partner because I like him a lot. I think it is necessary to tell him about my health. Just like protecting my family, I want to protect him too. I have already been infected with HIV. I do not want to harm others. Because I like him, I am obligated to tell him and protect him. So, I choose to tell him”*.(participant #5, 35 years old)

##### Being Forced to Disclose the HIV Status

The participants’ families discovered that they took the antiretroviral medications and began inquiring about it. Therefore, the participants had no choice but to disclose the HIV status. Some PLWH are forced to inform others of their HIV status because others maliciously spread information about their condition or Internet friends unexpectedly exposed their HIV status. Some HIV status disclosures happened when they were at hospital with their partners for checkups. The participants described their experiences as follows:

*“One of the antiretroviral medications needs refrigeration. One time my dad and stepmom saw it because I did not hide the medication well. They questioned me about the medication. Because they both knew about my sexual orientation, they directly associated “a gay taking medications” with “HIV”*.(participant #14, 33 years old)

*“The day I broke up with my ex-boyfriend, I quarreled with him. As a result, he was shouting outside our rental house. So the neighbors knew that I am an HIV- positive person. He also went to my workplace and told my colleagues that I am HIV positive. I was very embarrassed. Later, some of his friends and others asked me if I was infected with HIV”*.(participant #6, 25 years old)

#### 3.2.2. Experiences after Disclosure

After the participants voluntarily or involuntarily disclosed their HIV status to families, friends, colleagues, employers, or healthcare providers, they began to receive differential treatments such as getting unwanted attention in school, being excluded at the workplace, being isolated at home for the daily basic tasks and being treated differently when seeking medical care. Some participants found ways for stress relief, re-evaluated their perceptions toward the illness, and finally, learned to treat their HIV infection with a calmer mentality. Some participants used their personal experiences to assist others to overcome the hardship. This phase encompassed three themes, as presented below:

##### Receiving Special Considerations and Requirements from School or Work

When the participants were still students, schools provided counseling, limited their participation in specific activities, and hoped that they would initiate the request to quit schools. One participant described his experience as follows:

*“At first, my school cared a lot about my health. Then, they became more concerned about whether I would choose to quit school. I was told that they would notify my parents if I did not cooperate. I was asked to wash dishes and clothes separately from others. I was not allowed to swim. To me, quitting school created an excruciating scar that I could never forget. It was like someone cut my face with a knife”*.(participant #19, 26 years old)

PLWH experienced difficulties in finding and keeping employment. The most common issues were discriminatory treatment from colleagues and employers, and termination of employment after HIV status disclosure. One participant described his experience as follows:

*“I took Efavirenz and had severe rashes. My boss found out about it. He became concerned about it and asked me to see a doctor. My rashes were rather severe at the time. Because my work was about organic food, we had heightened sensitivity about medicine and diseases. So, my boss kept on questioning about my rashes. He even requested information about my doctor’s visit and medications so that he might get a second opinion. He believed that he had the right to know because he simply wanted to help a colleague and he was my boss. In the end, he even threatened to fire me if I did to provide the requested information. This had jeopardized my right”*.(participant #14, 33 years old)

##### Receiving Differential Treatments in Life and When Seeking Medical Care

After the participants’ families became aware of the HIV infection, the participants were excluded from living in the same household with family members, had to face family and friend’s finger-pointing, became isolated, and were evicted from home. Due to family members’ misunderstandings about the illness and fear of being infected, the participants’ personal items were separated from other family members’ belongings. These treatments made the participants feel excluded from their family. The participants described their experiences as follows:

*“As far as the tableware, my dad bought bowls, chopsticks, spoons, and other silverware for my personal use. So, I told my mom I will not eat at home. I thought … why would you exclude me! I wanted to say how could it become like this. I realized, then, I was different from others”*.(participant #7, 27 years old)

*“I was isolated from the beginning. My parents thought HIV was transmitted through droplets. So, my dad separated my toothbrush and toothpaste from other family member’s personal care items. The washbasin went from one to two. Originally, we only had one washbasin in the bathroom. My dad added another one. I thought he has gone overboard. Our bathroom was rather small. Why complicate things! At that moment, I began to feel sad about the situation”*.(participant #7, 27 years old)

After the participants voluntarily disclosed their HIV status to friends and partners, they became excluded from their peers. Close friends purposefully distanced from them, avoided HIV-related topics, and eventually terminated the friendships. One participant described his experience as follows:

*“After my boyfriend knew about my HIV infection, he chose to be silent and disappeared for a while. He appeared one day and wanted to break up with me. I held him and asked him not to leave me. He said, don’t force him. He could not do it. My ex-boyfriend and I are still deeply in love with each other. But our relationship could not continue because of my HIV infection”*.(participant #13, 30 years old)

After confirmation of the diagnosis, PLWH had to regularly check the CD4 count and viral load and attend to their health needs. When they voluntarily informed healthcare providers about the HIV status, they experienced unfriendly verbal and physical treatments from the healthcare providers. Healthcare providers frequently refused to see them. Additionally, the implementation of the national PharmaCloud System might contribute to differential treatments toward the PLWH in Taiwan. To facilitate patient safety and reduce duplicate prescriptions, the National Health Insurance Administration implemented the PharmaCloud System and Cloud patient record system. However, PLWH were concerned that these Cloud systems might expose their HIV status and negatively impact their right to receive healthcare. The participants described their experiences as follows:

*“When I took my lab results to my primary care physician, I could tell that he knew about my HIV infection. He was taken back by my HIV status. I was afraid already. But it was hurtful having to face my physician’s reaction”*.(participant #15, 42 years old)


*“When the healthcare providers knew that they might have an HIV-positive patient, they became concerned about the possibility of being infected with HIV. They were not sure if they should be wearing full personal protective equipment. When I heard about their reactions, I did not want to share my HIV status with them or my social work friends”.*
(participant #16, 26 years old)

##### Stress Relief and Restart

The HIV-related stigma made a significant impact on the PLWH. After HIV status disclosure, some participants were relieved from the emotional stress and received support from friends and families. Some participants adjusted their perceptions toward HIV infection, developed their personal views about the illness, and shielded themselves from the outside influences. The participants described their experiences as follows:

*“I think (after I told others about my HIV status) my emotional stress had decreased. If you did not say anything about it, you would not have the freedom to do things. But, after you told others, they would pay attention to my illness and I would also take measures to prevent the spreading of the illness. You would feel less stressed”*.(participant #5, 35 years old)

*“Watching TV, listening to music, avoid talking about it, and then exercising. I think time helped getting rid of these unpleasant feelings. It might be hidden somewhere in my heart. I try not to think about it”*.(participant #10, 31 years old)

PLWH repeatedly hesitated about the decision of HIV status disclosure. After they disclosed the illness, they might encounter positive or negative feedback and treatments. Some PLWH drew from their past experiences to assist other PLWH in facing the illness. From the experiences of assisting others, they found confidence and returned to their lives. One participant described his experience as follows:

*“I was not confident. However, I have been working hard to identify an aspect to restore my confidence, such as work. I wanted to do a good job with work and live a good life so that my friends and families would not worry about me. That would be my source of confidence and happiness”*.(participant #10, 31 years old)

## 4. Discussion

### 4.1. Phase One: Experiences before Disclosure

In this study, interviews of 16 participants found that the PLWH were “*torn between fear of unemployment/isolation and desire to protect closed ones*” before HIV status disclosure, which is consistent with results from studies conducted in Taiwan [21], China [20], the Netherlands [36], and the United States [23]. Multi-faceted concerns regarding the decision of disclosure have been reported in the literature, including hostility from the social environment [23,37], family disappointment [20], and discriminatory medical treatment [21,27,36,38]. Stutterheim et al. [36] found that it is particularly challenging for the HIV-positive healthcare providers to disclose the HIV status in workplaces. The HIV-positive healthcare providers concealed their HIV status due to the fears of negative reactions, previous experiences of negativity, and recommendations to conceal the HIV status [36]. HIV status disclosure should be a carefully thought-out decision due to its multi-faceted impact on the lives of PLWH. Additionally, the purpose, timing, location, and conditions for HIV status disclosure should be the choice of the PLWH [39]. It is critical for healthcare professionals to assist PLWH in assessing the purpose, timing, location, and conditions for HIV status disclosure. 

### 4.2. Phase Two: Experiences after Disclosure

In this study, the participants disclosed the HIV status to partners, friends, and family members. The participants’ order of priority for disclosure to family members were partners, mothers/sisters, brothers, and fathers. This order of disclosure may be attributed to the impact of the patriarchal society in Taiwan where a father is considered an authority figure. PLWH may view disclosure to their fathers (as authority figures) as the last resort because of the perceived difficulty of accepting the HIV-positive status of their children. On the other hand, the participants’ closer relationships with mothers, sisters, and brothers made the disclosure process easier. In this study, the participants disclosed the HIV status to the most trusted family members and colleagues in case they needed assistance for unexpected incidents, which is consistent with the literature indicating that levels of trust and closeness of relationships as the main predictors for HIV status disclosure [20,38]. Additionally, all participants voluntarily disclosed their HIV status to partners due to the desire to protect them from HIV infection. Specifically, 68.42% of the participants who disclosed their HIV status to their partners had reached undetectable viral load which signifies untransmissible HIV infection [40,41]. This result shows that the decision of disclosure may be affected by the levels of viral load. The PLHW’s willingness of disclosing HIV status to partners may increase if the PLWH closely adhered to the treatment to control the viral load at an undetectable level.

In this study, 31.58% of the participants choose not to disclose HIV status to family members due to the fear of unexpected negative consequences after disclosure. The participants who disclosed their HIV status to family members reported differential treatments, unexpected negative reactions, and stigmatization from their family members, which is consistent with the literature [15,18,19,24]. The challenge of disclosing HIV status to family members may be explained by the three-layered Stigma Model for China by Yang and Kleinman [24] and the Chinese Confucius philosophical beliefs. The Stigma Model for China illustrates the multi-faceted and pervasive impact of stigma (in this case, HIV-positive status) of three aspects: societal aspects, moral aspects, and subjective, collective, and interpersonal aspects. In other words, the stigma resulting from the disclosure of HIV status may have a significant impact on the entire family, social networks, and interpersonal relationships [24]. Additionally, the process of HIV status disclosure among PLWH in Taiwan may differ from the process of PLWH in Western society due to the influence of the Confucius philosophical beliefs, which place a strong emphasis on moral characters, harmonious relationships, and family. In this study, all of the participants were MSM, which presented additional challenges in the HIV status disclosure. As they contemplated the decision to disclose the HIV status, they faced pressures from having to expose their identities as MSM and PLWH and feared the possible resulting stigmatization. In Smith et al.’s study, the MSM PLWH reported experiencing mental and emotional distress, social segregation, social withdrawal, rejection in relationships, high-risk behaviors, alcohol consumption, drug use, hiding or selectively disclosure of HIV status, and poor medication adherence as consequences of HIV-related stigma [15].

All of the participants in this study were MSM. However, the experiences of HIV status disclosure of female PLWH also require attention. Women reported higher levels of stigma and religious well-being than men [23]. Therefore, it is important to understand the experiences of HIV status disclosure in female PLWH since gender impacts motivation of disclosure and responses toward the disclosure [42]. HIV status disclosure was an easier process for Chinese males than for females, which highlights gender inequality in the Chinese culture [20]. Fekete, Williams, Skinta, and Bogusch [43] discovered that females were more likely to have regrets after disclosure of HIV status. The female PLWH had greater concerns about stigmatization after HIV status disclosure than males. Additionally, they worried about the consequences after seeking assistance from their own social networks. The results showed a negative correlation between quality of life and disclosure concerns [43]. In this study, all of the participants were male PLWH. During recruitment, most female PLWH held a conservative attitude and refused to participate in the study. Therefore, we were unable to explore the experiences of HIV status disclosure in female PLWH.

In this study, 13 participants reported “*differential treatment in life and when seeking medical care*” after disclosing their HIV status. Results from the 2017 Taiwan HIV Stigma and Discrimination survey revealed that 7.3% of the HIV positive individuals were refused medical services due to their HIV status (including dental care), and 10.6% of them had their HIV status exposed by healthcare professionals without their permission [21]. Ting, Li, and Fang discovered that 84.6% of surgeons were concerned about contracting HIV from surgical operations [38]. Results of a survey by the Taiwan Lourdes Association [44] found that 70% of the PLWH worried about unemployment after HIV status disclosure, and refusal or differential medical care because of implementation of the PharmaCloud System. Fifty of the PLWH were afraid of disclosing HIV status to healthcare professionals when seeking medical care because of the fear of receiving differential treatments. These findings revealed the significant impact of HIV-related stigma on PLWH as they seek healthcare in Taiwan. It is critical for healthcare professionals to evaluate their own attitudes toward PLWH since biased and discriminatory perceptions toward PLWH may transform into actions in practice and contribute to stigmatization [13].

### 4.3. Limitations

This study used a purposive sampling method for recruitment. All of the participants in this study were MSM. In Taiwan, the ratio of male/female PLWH is 12:1. Therefore, access to the female PLWH was limited. Additionally, the primary reasons for HIV infection in the Taiwanese female PLWH are heterosexual unsafe sex and intravenous drug use (IDU) [2]. Female PLWH who contracted HIV infection from spouses dislike inquiries about their HIV status and often sought medical care with the spouse. Therefore, they may refuse to participate due to the sensitive nature of HIV infection. Additionally, the female PLWH who are IDU may be concerned about the legal implications and refused to participate. Future studies are recommended to recruit female PLWH to gain a comprehensive understanding of the process of HIV status disclosure. Additionally, we recommend using the mixed methods approach for a more synergistic analysis and utilization of the qualitative and quantitative data to gain a comprehensive and authentic understanding of the experiences of stigmatization among the PLWH in Taiwan.

## 5. Conclusions

The findings of this study revealed two phases and six themes regarding HIV status disclosure and HIV stigmatization. Disclosure of HIV status is a process that requires careful considerations before informing family members, friends, partners, and healthcare professionals. In this study, the PLWH struggled with the decision of disclosure. Both disclosure and concealment of HIV status are associated with pressures and risks. It is essential for healthcare professionals to assess the social networks and family structures/situations of PLWH, and assist them in considering methods of disclosure and possible situations/scenarios before and after disclosure to avoid HIV-related stigmatization after disclosure.

Healthcare professionals are recommended to assess the PLWH’s experiences after the HIV diagnosis and tailor health education to meet their needs. To assist with the PLWH’s HIV status disclosure to family members, healthcare professionals need to carefully evaluate their degree of negative self-image, family structures and support, and provide individualized guidance. Collaborative care from interdisciplinary teams (such as physicians, nurses, psychologists, social workers, pharmacists, dietitians, and medical laboratory scientists) is crucial in meeting their multi-dimensional needs.

## Figures and Tables

**Figure 1 ijerph-18-05089-f001:**
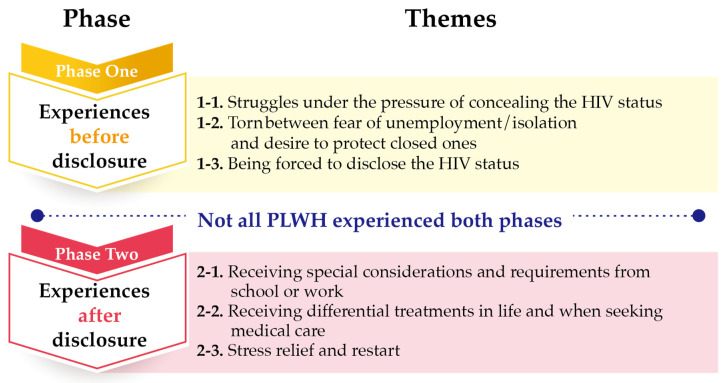
Summary of phases and themes.

**Table 1 ijerph-18-05089-t001:** Interview guide.

1. Whom did you tell when you learned that you are infected with HIV? Why did you tell him/her? Who else (family, partner, friends, or colleagues) knows about this? How did they know about this?
2. How do people in your social circle (friends, classmates, colleagues, or family) view people living with HIV? How do they perceive people living with HIV? How do you feel about this?
3. What did you consider when you decided to tell your family about your diagnosis? Why did you conceal your HIV infection? What do they think about your HIV infection?
4. How would you describe your interactions with your family after they were told about your diagnosis? What has changed? How do the changes impact you?
5. How would you describe your interactions with your relatives, neighbors, and friends after they were told about your diagnosis? How does this impact you?
6. How would you describe your interactions with your colleagues at work after they were told about your diagnosis? How does this impact you?
7. What do you think about stigmatization toward people living with HIV? How do your friends think about this? How does this make you feel?
8. Could you talk about the factors you considered when selecting a hospital for medical attention? Would you take the initiative to inform healthcare professionals about your HIV infection? What factors play into your decision to inform the healthcare professionals about your HIV infection?
9. Has your diagnosis been unexpectedly exposed? Could you talk about the situation? How did it make you feel?

## Data Availability

The data presented in this study are available on request from the corresponding author. The data are not publicly available due to confidentiality.

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
