# Peer review of "The Lived Experiences of Stigmatization in the Process of HIV Status Disclosure among People Living with HIV in Taiwan"

_ijerph, 2021, doi:10.3390/ijerph18105089_

Round 1

Reviewer 1 Report

The article is well written and scientifically correct.
The bibliography must be redone because it does not fully comply with the requirements of the journal in which the article will be published.

Congratulations to the authors for addressing an important public health issue.

Author Response

Thank you so much for the opportunity to revise this manuscript. We truly appreciate the reviewer’s comments. The bibliography has been careful reviewed and revised.

Reviewer 2 Report

The study is interesting. However, although it is located under the qualitative perspective, I consider it pertinent to indicate that it would be necessary to complement it with a quantitative vision. It is understandable that there is a directed and reduced sample, which is not limiting to include some analyzes such as contingency tables (Zar, J. H. (2010). Biostatistical analysis. 5th ed., Pearson Prentice-Hall, Upper Saddle River, New Jersey.), to compare between subgroups of men based on the responses they issued from systematizing the Interview guide with some scoring scale (Likert, for example, or another to assign a number). The authors can still apply this approach, which would allow them to quantify and expose frequencies in percentages of the content analysis of the record of the interviews with the participants.

Regarding the discussion, I consider it very extensive and there could be a style adjustment, for example, between the line 354-369, it can be adjusted as a single sentence that indicates a specific way that the results of the authors coincide with the study in Holland and the fears of disclosing the aerological status due to the expected negative reactions.

In the Limitations section, I suggest including a sentence stating that for the extrapolation of the results, it would be convenient to carry out studies with a mixed qualitative and quantitative methodology to be more forceful with the results, thus contributing to the reproducibility of future research.

In general, the research addresses one of the issues that is affecting individuals with this type of disease.

Author Response

Thank you for the opportunity to re-submit our manuscript, titled “The lived experiences of stigmatization in the process of HIV status disclosure among people living with HIV in Taiwan”. We have carefully read the reviewers’ comments and made revisions accordingly. Our response to the reviewers’ comments is attached, with the revised portion highlighted in red. Please contact me if you have questions or need additional information. Thank you very much.

Reviewer 3 Report

I think that articles like this are needed. Reading about the reality of another country and how stigmatized HIV-positive MSM are was shocking but really interesting. I think that a thorough English revision is needed. I also have a few comments:

  • I would expand a little bit on how Taiwanese culture condemns MSM on lines 58-60. I would suggest just to explicit better in a phrase the link between the Taiwanese motto reported and the negative attitudes towards MSM.
  • Also, MSM is treated like singular sometimes in the text, but it refers to mEn. (see line 59)
  • Some information is repeated twice and result redundant. E.g. the interview was transcribed verbatim within 24-72 hours after completing the interviews
  • “All participants were single biological male aged between 22 and 42 years old”. What the authors mean by single? Without a relationship?
  • I would expand a bit more on the clinical implications of the study.

Author Response

(The authors gave the same response as above.)

Reviewer 4 Report

This is a qualitative study exploring the experiences of PLWH in Taiwan. The authors identify different themes relevant to before and after disclosure of HIV status to partners, family, and friends. While this study could potentially be very valuable, it is difficult to evaluate the importance or value of the findings given the current presentation. The manuscript suffers from confusing language and poor organization and lacks a clear motivation for collecting these interviews.

Confusing language

Throughout the manuscript, there are examples of shaky English grammar and imprecise language. This is particularly problematic where cultural explanations and the reporting of results implies over-generalization. For example, the authors cannot possibly know what all “Taiwanese believe” (line 57) or that the results show that PLWH (rather than some men in the study) “put themselves and their families under varying degrees of stress” (line 178-9).

There are also a few examples of oddly translated phrases. This is particularly confusion in the explanation of “loss of face” (paragraph at 83), where face is used as a concept but not adequately defined. Though I am not familiar with the Chinese concept described here, it would seem that the concept does not translate well to “face” without reference to “loss of” or “saving.” The cultural explanations are interesting and valuable to an international readership, but they must be more thoroughly explained.

Poor Organization

Organization can be improved throughout, but especially in the presentation of the themes, both in the abstract and the results, and in the discussion. The notion of “two phases” each associated with three themes is unnecessarily confusing. In the abstract, the phases should be named, but it might be clearer to simply state that conversations focused on experiences before and after disclosure. Currently, the abstract refers to 2 phases without saying what they are.

The discussion is confusing because it contains many elements that I would expect to find in other parts of the paper. The review of previous work on PLWH in Taiwan, China, and elsewhere would traditionally be presented in the background section. Results of the current study, such as reported behaviors of health professionals, should appear in the results. This all makes it difficult to understand the contribution of the paper.

Lack of Motivation for the Study

There is something intrinsically compelling about these heartbreaking accounts of stigma and the accompanying explanations of Taiwanese culture. However, the manuscript could be improved by explaining why the study was important enough to undertake, what the authors hoped to learn, and more clearly connecting the contribution to the existing literature by highlighting what is new or novel about the study (not just how the study is consistent with current literature). Given the relatively small population of PLWH in Taiwan (is it truly >2000?), why is this study important? If the study is truly meant to shed light on prejudice among healthcare workers, this should be an organizing principle of the manuscript.

Author Response

(The authors gave the same response as above.)

Round 2

Reviewer 3 Report

I think that the manuscript really improved after the thorough revision the authors made. I do not have further comments.

Author Response

(The authors gave the same response as above.)

Reviewer 4 Report

I have made editorial suggestions in the attached PDF. I suggest using a grammar check in a word processor before publication. 

Author Response

(The authors gave the same response as above.)
